# Foliar Diseases and the Associated Fungi in Rice Cultivated in Kenya

**DOI:** 10.3390/plants11091264

**Published:** 2022-05-07

**Authors:** Everlyne M. Nganga, Martina Kyallo, Philemon Orwa, Felix Rotich, Emily Gichuhi, John M. Kimani, David Mwongera, Bernice Waweru, Phoebe Sikuku, David M. Musyimi, Samuel K. Mutiga, Cathrine Ziyomo, Rosemary Murori, Lusike Wasilwa, James C. Correll, Nicholas J. Talbot

**Affiliations:** 1Department of Botany, School of Physical and Biological Sciences, Maseno University, Kisumu P.O. Box 3275-40100, Kenya; mumbuaeve@gmail.com (E.M.N.); phoebenyangi@yahoo.com (P.S.); davidmusyimi2002@yahoo.com (D.M.M.); 2Biosciences Eastern and Central Africa-International Livestock Research Institute (BecA-ILRI) Hub, ILRI Complex, Old Naivasha Road, Nairobi P.O. Box 30709-00100, Kenya; m.kyalo@cgiar.org (M.K.); b.waweru@cgiar.org (B.W.); c.ziyomo@cgiar.org (C.Z.); 3Department of Water and Agricultural Resource Management, School of Agriculture, University of Embu, Embu P.O. Box 6-60100, Kenya; philemonsorwa22@gmail.com (P.O.); rotich.felix@embuni.ac.ke (F.R.); 4Kenya Agricultural and Livestock Research Organization, Kaptagat Road, Loresho, Nairobi P.O. Box 57811-00200, Kenya; gichuhiemily@gmail.com (E.G.); kimanijm69@gmail.com (J.M.K.); dmwongerathuranira@gmail.com (D.M.); lwasilwa@gmail.com (L.W.); 5Eastern and Southern Region Office, International Rice Research Institute, ILRI Complex, Old Naivasha Road, Nairobi P.O. Box 30709-00100, Kenya; jcorrell@uada.edu; 6Department of Entomology and Plant Pathology, Division of Agriculture, The University of Arkansas System, Fayetteville, AR 72701, USA; r.murori@irri.org; 7The Sainsbury Laboratory, Norwich Research Park, University of East Anglia, Norwich NR4 7UH, UK; nick.talbot@tsl.ac.uk

**Keywords:** rice, disease surveillance, multi-infections, foliar diseases, blast, brown spot

## Abstract

We conducted a survey to assess the occurrence and severity of rice blast and brown spot diseases on popular cultivars grown in the Busia, Kirinyaga, and Kisumu counties of Kenya in 2019. Working with agricultural extension workers within rice production areas, we interviewed farmers (*n* = 89) regarding their preferred cultivars and their awareness of blast disease, as this was the major focus of our research. We scored the symptoms of blast and brown spot and assessed the lodging, plant height, and maturity of the crops (days after planting). Furthermore, we collected leaf and neck tissues for the assessment of the prevailing fungal populations. We used specific DNA primers to screen for the prevalence of the causal pathogens of blast, *Magnaporthe oryzae*, and brown spot, *Cochliobolus miyabeanus*, on asymptomatic and symptomatic leaf samples. We also conducted fungal isolations and PCR-sequencing to identify the fungal species in these tissues. Busia and Kisumu had a higher diversity of cultivars compared to Kirinyaga. The aromatic Pishori (NIBAM 11) was preferred and widely grown for commercial purposes in Kirinyaga, where 86% of Kenyan rice is produced. NIBAM108 (IR2793-80-1) and BW196 (NIBAM 109) were moderately resistant to blast, while NIBAM110 (ITA310) and Vietnam were susceptible. All the cultivars were susceptible to brown spot except for KEH10005 (Arize Tej Gold), a commercial hybrid cultivar. We also identified diverse pathogenic and non-pathogenic fungi, with a high incidence of *Nigrospora oryzae*, in the rice fields of Kirinyaga. There was a marginal correlation between disease severity/incidence and the occurrence of causal pathogens. This study provides evidence of the need to strengthen pathogen surveillance through retraining agricultural extension agents and to breed for blast and brown spot resistance in popular rice cultivars in Kenya.

## 1. Introduction

Rice (*Oryza sativa* L.) provides 20% of the global dietary energy used by humankind and is, therefore, one of the most important cereal crops. The demand for rice in Africa is growing at a rate of 6%, which is faster than any other region in the world. With 25% (US$ 6 billion) global net imports, Africa was ranked the second largest importer of rice in 2018 [1]. Kenyan rice per capita consumption increased from 12.7 to 20.6 kg between 2008 and 2018 due to an increase in the population and changes in eating habits [2]. The current annual national rice consumption in Kenya is about 949,000 metric tons, compared to its annual production of 180,000 metric tons (IRRI Kenya https://www.irri.org/where-we-work/countries/kenya; accessed on 27 March 2022). 

In Kenya, 95% of rice is produced with government support, through the provision and management of water by the National Irrigation Authority (NIA) in designated production areas, called rice irrigation schemes, while the remaining 5% of the rice is produced through rain-fed agriculture [3]. There are four rice irrigation schemes under the NIA, including Mwea in Kirinyaga County, Ahero and West Kano in Kisumu County, and Bunyala in Busia County [4]. Production is mainly carried out by smallholder farmers who are allocated small portions of land (approximately 0.2 ha), which are converted into paddies [1]. The farming systems practiced by these farmers vary widely, with the majority being constrained by low input, drought, pests, and diseases. Several foliar diseases affect rice, the most common being rice blast, brown spot, rice yellow mottle virus, bacterial leaf blight, leaf streak, sheath rot, and the tungro disease complex [5]. 

The most common rice fungal diseases include blast (*Magnaporthe oryzae*), brown spot (*Cochliobolus miyabeanus*), sheath rot (*Sarocladium oryzae*), root rot (*Fusarium moniliforme*), and seedling blight (*C. carbonum* and *M. oryzae*) [6]. Among the foliar diseases, rice blast is the most damaging, causing severe grain yield reduction in upland and lowland rice-growing regions in Kenya [7]. The blast fungus causes damage to rice plants at all stages of growth, leading to characteristic oval—diamond leaf lesions, which later turn whitish-to-gray at the center, with brown margins. The collars appear brownish and the attached leaves die back in heavy infections. Later, the stem nodes appear darkish brown and rotten and the panicles above the neck die back [8]. In sub-Saharan Africa, rice blast causes grain yield losses of up to 100% [9]. Brown spot disease is characterized by circular or oval brown lesions, which reveal halos on the leaves during the tillering stage. The lesions appear grayish at the center and reddish-to-brownish at the margin, while the stems are discolored [10]. Brown spot is endemic and is mainly associated with low-input agriculture and crops experiencing physiological stress. The estimated yield losses due to brown spot are in the range of 10%, although accurate estimates are lacking [11,12]. Losses due to brown spot result from reductions in the leaf area index, the early senescence of diseased plants, reductions in the number of tillers, and reductions in root and shoot elongation [13]. Although it is known that farmers use cultural methods, such as clean beds and fungicides, to control these diseases, the frequency of the use of chemical pesticides by small-scale farmers could be low due to high costs, and it is not known whether these chemicals are applied at the appropriate time and/or rate [14]. 

Plant tissues contains a diversity of microorganisms that can play roles in growth and fitness or proliferate without causing harm or benefits to host plants. Several studies aiming to determine the diversity of fungi in rice tissues have reported diverse microbiomes associated with rice growth [15,16,17,18]. Among the fungi found in rice tissue, *Aspergillus*, *Chaetomium*, *Fusarium*, *Penicillium* and *Trichoderma* species have all been reported [18,19]. A study aiming to determine the fungal communities on rice in Thailand identified a total of 22 and 31 fungal species in the leaves and stems, respectively [20]. Knowledge of the fungal communities associated with foliar diseases in rice is important because it can help researchers to understand the influence of these fungi on the overall disease impact and their potential uses for the biocontrol of blast disease. The effective management and control of rice diseases requires regular pathogen surveillance to establish the occurrence and changes in fungal populations over time. Pathogen surveys can be achieved through field visits for visual assessments, interviews with farmers, and the use of questionnaires to collect and collate relevant information during field visits. Both symptomatic and asymptomatic leaf and neck samples can also be collected to identify causal pathogens and/or endophytes. 

This study was part of a project that was primarily focused on breeding for rice blast resistance as the most sustainable method for disease management. In this study, the identification of the fungal species associated with blast and brown spot disease was carried out in rice fields in major production areas across Kenya. The study focused on the dynamics, incidence, and severity of blast and brown leaf spot disease on farmer-preferred rice cultivars. We identified the fungal species associated with the most significant foliar diseases under field conditions. The study complements ongoing efforts to identify farmers’ favored cultivars, evaluate their field performance, and obtain insights into the complexity of foliar diseases and diagnostic challenges in Kenya. We provide new fundamental information regarding the major foliar diseases of rice in Kenyan production systems and how this knowledge can be utilized in an effective disease management program. 

## 2. Results

### 2.1. Frequency and Agronomic Characteristics of Popular Rice Cultivars 

We conducted a survey on the popularity of cultivars among farmers and the occurrence of two main foliar diseases in small-holder farms within the Busia, Kisumu, and Kirinyaga counties of Kenya. The rice production system (based on the source of water, and whether the crop is rainfed or irrigated for crop growth) is mostly irrigation, using water that is provided by NIA, but a smaller proportion of the crop is cultivated under rainfed conditions. Farmers typically grow rice in small land parcels ranging from 0.04 to 4.05 ha (Table 1). The number of cultivars among the farms sampled in each county were as follows: Busia (*n* = 11), Kisumu (*n* = 7), and Kirinyaga (*n* = 3). NIBAM11 (commonly known as Pishori) and IR05N221 (also known as Komboka) are aromatic Indica rice cultivars and are cultivated by farmers in the three counties (Table 2). The most popular cultivars for each region were as follows: NIBAM11, at 92% in Kirinyaga, Vietnam, up to 17% in Busia, and NIBAM108 (also known as IR2793-80-1), at 32% in Kisumu. The NIBAM11 cultivar (cv.) was mainly preferred for its aroma and high market value, the Vietnam cv. For its high grain yield, and NIBAM108 cv. for its resistance to multiple diseases and pests (Table 2). Although hybrid cultivars KEH10004 (also called Arize 6444) and KEH10005 (also called Arize Tej Gold) are presumed to yield more grain than the other cultivars, farmers did not associate these cultivars with the key market trait of being aromatic, and their frequencies were therefore low in Busia (KEH10004, 3%; KEH10005, 7%) and Kisumu (KEH10005, 7%). Interestingly, farmers from Busia and Kisumu adopted six cultivars (including Abednego, Nyaboda, Pakistan K23, Palata, Supa 1, and Vietnam) which were not registered in Kenya and whose seed origin was claimed to be Uganda (Table 2).

### 2.2. Incidence of Blast and Brown Spot Diseases and a Confirmation of Infection by the Causal Fungi

Across the seventy-two rice plots (farms), where the survey was conducted in Busia and Kisumu counties, there was a moderate blast incidence of 42% (30/72). However, the brown spot incidence was higher, at 60% (43/72). The frequencies of the rice blast symptoms across the rice production systems were as follows: the highest incidence of blast was observed in the rain-fed farms in Busia (49%), followed by the irrigated farms in Busia (31%), and then the irrigated farms of Kisumu (17%) (Figure 1). Similarly, the highest brown spot frequencies were observed in Busia under rain-fed (42%) and irrigated (42%) conditions, while the frequency was lowest in the irrigated fields of Kisumu (16%) (Figure 1). Despite the presence of moderate incidences of the two diseases, there was a low isolation frequency for *M. oryzae* and *C. miyabeanus*. Two isolates of *M. oryzae* were obtained from neck tissues from Busia and Kirinyaga. No isolates of *M. oryzae* were obtained from the leaf samples. Three isolates of *C. miyabeanus*, two from Busia, and one from Kisumu were isolated from leaves showing brown spot symptoms.

We used specific PCR primers to test for the presence of *M. oryzae* and *C. miyabeanus* in asymptomatic and symptomatic leaf samples. Based on a likelihood-ratio chi-squared test of the PCR results, the proportion samples with detectable *M. oryzae* (16/29 or 55%) in the asymptomatic group did not differ significantly (*X*^2^ (1, *n* = 72) = 0.021, *p* = 0.8881) from those in the symptomatic category (23/43; 59%). For brown spot, the symptomatic samples had a statistically (*X*^2^ (1, *n* = 72) = 12.78, *p* = 0.0004) higher proportion of leaves (21/43; 49%) with detectable *C. miyabeanus* compared to the asymptomatic category (3/29; 10%).

Based on the field scores, a strong correlation between blast disease severity and incidence (*r =* 0.96, *p* < 0.0001) was observed. However, the infection rate, that is, the proportion of samples with PCR-detectable pathogen, was not correlated with either the incidence or the severity of blast (Appendix A). Further, we tested for an association between the blast disease magnitude and the presence of the causal pathogen detected by PCR for *M. oryzae* or *C. miyabeanus*. We found that the samples with detectable *M. oryzae* had a higher disease severity (3.3) than those without a detectable pathogen (1.5; *p* = 0.0129). Based on field assessments, the symptoms of the two diseases co-occurred in 41% of the surveyed farms, while co-infections by the two pathogens based on PCR-based detection were found to be 20%.

The incidence and severity of blast were significantly associated with awareness of the disease (*p* = 0.0004, Table 3). Although the rice production systems did not differ statistically in terms of incidence or severity of blast, the proportion of farmers who were aware of blast disease was higher under irrigated paddy schemes (33%) than under the rain-fed system (20%) (Figure 2). The mean incidence of rice blast was eight times higher in farms where farmers were unaware of the disease (Figure 1). Similarly, farmers who were unaware of blast were from farms with a higher mean disease severity (4) than those whose farmers were aware (0.3) of the disease (Figure 3). The cultivar effect was marginal for the rice blast incidence, but strong for disease severity (Table 3). In terms of the incidence and severity of blast, the most susceptible cultivars were NIBAM110, Vietnam, and NIBAM108, while the most resistant were NIBAM109 and NIBAM11, as shown in Table 3. Based on the PCR results for the presence of the pathogen, the cultivars with the lowest rate of infection by *M. oryzae* were: Palata (*n* = 1, 0%), Pakistan K23 (*n* = 3, 33%), and NIBAM108 (*n* = 14, 36%), while the most infected were NIBAM110 (*n* = 10, 80%), NIBAM109 (*n* = 5, 80%), NIBAM11 (*n* = 10, 50%), and Vietnam (*n* = 7, 71%) (Table 4). For brown spot, the least infected cultivar was KEH10005 (*n* = 6, 33%), while the rest were susceptible, with an infection rate of over 50% (Table 4).

### 2.3. Relationship between Disease Components and Agronomic Traits of Rice

We used Spearman’s non-parametric correlation tests to assess the associations between the three agronomic traits (i.e., the age of the crop, the plant height and the lodging), and the incidence of the two diseases. We found no significant correlation (*p* > 0.05) between any of the agronomic traits and the occurrence of blast. By contrast, the brown spot incidence was positively correlated with the plant height. Furthermore, the brown spot incidence had a marginal positive correlation with lodging (Appendix A).

### 2.4. Fungal Profiles in Leaf and Neck Tissues of Rice

We plated the leaf and neck tissues for the isolation of culturable fungal species. All the plated neck tissues had characteristic rot symptoms. Most of the plated leaves (60%) showed symptomatic necrosis of varying lesion sizes and types. Owing to the typical nature of the culturing of microorganisms, in which some samples do not show any growth, we collected data on the isolates obtained from the leaves or neck tissues from each region, and not samples that were plated. The numbers of isolates that were successfully sequenced and identified across the samples from each of the counties were as follows: Kirinyaga (leaves, *n* = 48; neck, *n* = 57), Busia (leaves, *n* = 20; neck, *n* = 29), and Kisumu (leaves, *n* = 10; neck, *n* = 14). The sequences of the isolates were deposited into the National Center for Biotechnology Information (NCBI) gene bank in a file SUB10723570 with accession numbers ranging from OM899814 to OM899972. 

Based on a comparison of the fungal sequences with those in the NCBI database, the isolates belonged to 17 genera (Figure 4). The three most frequent genera were *Nigrospora* (42%), *Eppicoccum* (15%), and *Fusarium* (13%), while the least prevalent (<1%) were Eutypella, *Mucor*, *Microdochium*, and *Neurocrassa* (Figure 4). It should be noted that the internal transcribed spacer region is generally used in the barcoding of fungi and, although it is acceptable for genus and species identification, it may fall short of accuracy for some species [21,22]. We used the NCBI results and with a general reference to the published literature on the determination of the putative fungal species isolated in this study. For accessions of genera with bottlenecks in the identification of species using ITS, a side-by-side comparison of the results from NCBI and UNITE was performed (Appendix A). Generally, we observed a higher diversity of fungal species in the neck than in the leaf tissue (Figure 5 and Figure 6). For the sampled leaves, the occurrences across Kenyan counties were in the following order: Busia (*n* = 14), Kirinyaga (*n* = 7), and Kisumu (*n* = 6) (Figure 5). For the rice neck tissue, the order of occurrence of diverse species was as follows: Kirinyaga (*n* = 18), Busia (*n* = 14), and Kisumu (*n* = 7) (Figure 6). Overall, forty-six fungal species were isolated, with the most frequent fungal species being *Nigrospora oryzae* (32%), *Epicoccum sorghinum* (15%), and *Fusarium equiseti* (11%). The rest of the species occurred at frequencies of less than 5%, with the expected causal organisms for common rice diseases, with *M. oryzae* being isolated at 3% in neck tissues from Busia (*n* = 2) and Mwea (*n* = 3), *C. miyabeanus*, at 2%, in leaves from Busia (*n* = 2) and Kisumu (*n* = 1), and one isolate of *Sarocladium oryzae* (sheath-rot fungi) in samples from Bunyala, in Busia county.

The frequencies of the most abundant fungal species were analyzed in the leaf and neck tissues. The most frequent fungal species in the rice leaves were in the following order of occurrence: *N. oryzae* (68% Kirinyaga), *F. equiseti* (23% Kisumu; 2% Kirinyaga), *E. sorghinum* (20% Busia; 17% Kirinyaga), and *C. miyabeanus* (12% Kisumu; 10% Busia). For the neck tissues, the most prevalent fungal species were as follows: *F. equiseti* (58% Kisumu; 17% Busia and 6% Kirinyaga), *N. oryzae* (43% Kirinyaga), and *E. sorghinum* (17% Kirinyaga; 11% Busia and 7% Kisumu). The fungal species that were isolated in both the leaves and the neck tissue were: *Cladosporium cladosporioides*, *E. sorghinum*, *F. equiseti*, *F. oxysporum*, *N. oryzae*, *N. sphaerica*, and *Pithomyces chartarum* (Figure 5 and Figure 6).

Owing to the high frequency of *N. oryzae* in the samples that originated from farmers who had complained of a blast-like epidemic in their fields, we conducted further analysis to gain insights as to whether the isolates were clones or diverse strains. The sequences of these isolates were aligned, and a neighbor-joining phylogenetic tree was generated based on Kimura’s two-parameter estimate model in CLC Genomics Workbench [23]. A comparison of the percentages of the nucleotide similarities showed that 92% of the sequences of the isolates were 100% similar, with the similarity in the rest ranging from 94% to 99.99% (Appendix A). Our efforts to induce sporulation were unsuccessful and, therefore, we did not confirm the cause of the blast-like symptoms using classical Koch’s postulates.

### 2.5. Relationship between Fungal Species and Rice Hosts

We obtained publicly available information regarding the relationship between the identified fungal species and the host. Based on previous reports, thirteen species isolated from either leaf or neck tissue were known to have caused at least fifteen diseases in rice and other cereals (Table 5). The foliar disease-causing fungi that were only isolated from leaves included *C. miyabeanus* (brown spot) and *Microdochium albescens* (leaf scald). The foliar pathogens isolated from only neck tissue were *M. oryzae* (blast) and *C. lunatus* (leaf spot), while *E. sorghinum* (leaf spot) was found in both neck and leaf tissue (Table 5). The root-rot-complex disease-causing fungus (*Lasiodiplodia theobromae*) was isolated from neck tissue, as were the seed-rot fungi (*Aspergillus flavus*, *Alternaria padwickii*, and *Fusarium incarnatum*). Other Fusarium species implicated in disease and rice-grain contamination were also isolated from the leaves (Table 5).

## 3. Discussion

In this study, we identified the rice cultivar preferences of farmers, recorded the incidence of blast and brown spot disease, and determined the fungal profiles in the neck and leaf tissues from rice collected in three major growing regions of Kenya. This information will inform future choices of cultivar panel for inclusion in subsequent studies to evaluate blast resistance in locally adapted rice germplasm collections. The direct interviews and field observations provided an opportunity to learn about rice production challenges and potential mitigation approaches in smallholder farming systems in Kenya. Rice production is mainly carried out in irrigated fields, and although the irrigation water is regulated, the crop is also affected by weather conditions. Thus, we acknowledge that the observations herein might have been affected by climatic information, and we did not collect the soil moisture data during the crop production seasons. Our findings from this study can therefore be considered a snapshot of the current situation, but one with key valuable and actionable insights, while the gaps in the study can be used as a foundation for further efforts to enhance rice production in Kenya.

The general findings of this survey reveal that farmers and agricultural extension agents cannot accurately identify diseases based on the observation of symptoms in the field. Although the original focus of our study was rice blast disease, our interactions with farmers and agricultural agents pointed to a need to also collect data on brown spot disease, because it is also prevalent in western Kenya and often occurs at the same time as blast, mirroring the importance of these diseases globally [5,7,41]. Alongside surveys, the plant pathologists in our team trained farmers and extension specialists in the identification of diseases based on symptoms to address this need. In a complementary activity, we collected samples for analysis using mycological and molecular techniques to complement the visual assessment of the diseases in the field. Our study, therefore, provided further evidence of the need to improve disease diagnostics and the training of Kenyan agricultural extension agents in plant disease identification, pathogen detection, and disease management, which are critically required if disease losses are to be mitigated in the region [1]. With a team of informed agricultural extension agents, farmers are able to obtain the most accurate information on the diseases affecting their fields and how to manage them. Although small-scale farmers may lack the funds with which to buy fungicides, knowledge of diseases could be helpful in the application of the appropriate control for a reasonable percentage of rice growers, particularly with support from government subsidies. 

There were more preferred cultivars in rain-fed and irrigated farms of the Western counties of Kenya (Busia and Kisumu) than in Central Kenya (Kirinyaga). Information on cultivar diversity enables breeders and seed dealers to plan for product development, distribution and the associated communication to different stakeholders across rice growing regions. The occurrence of multiple rice cultivars of lower commercial value in Busia and Kisumu, compared to the cultivation of one popular cultivar, NIBAM11, in Kirinyaga could be associated with a lag in rice commercialization due to a lack of knowledge and policy support for improvements in rice production, milling, and marketing in the two counties. Indeed, the farmers in Western Kenya reported that they sold un-milled rice to dealers from Uganda because they did not have efficient or affordable access to milling systems within their localities [42]. While planting diverse cultivars of varying disease resistance spectra could safeguard the crop against the virulent evolution of pathogens, grain yield and access to markets are equally, and perhaps more, important to farmers. We speculate that farmers in Western Kenya have adopted multiple foreign cultivars so that they can easily gain access to the Ugandan market for their rice. Although efforts to achieve mechanization and commercialization are ongoing in the main irrigation schemes under the National Irrigation Authority, Kirinyaga county has already received more support through multiagency teams and currently produces about 86% of the locally consumed rice in the region [43]. We conclude that there is still significant potential to enhance rice production to meet Kenyan consumption demands and satisfy regional markets based on cultivar selection [43]. Farmers in Western Kenya should be enlightened as to the availability and importance of certified rice seeds, as these can reduce the spread of some the major diseases within Kenya and across East Africa.

We observed differences in the responses of the popular rice cultivars to blast disease. Based on the disease severity scores and infection rates, the rice cultivars that were consistently susceptible to blast included NIBAM110 and Vietnam, while IR05N221 and NIBAM108 were moderately resistant. Although the cultivars Pakistan K23, Palata, and Abednego showed low disease scores, the sample sizes were also low and likely to have been insufficient to warrant any conclusive inferences. The cultivars NIBAM11 and NIBAM109 had low blast incidence and severity, but their infection rates were high. Previous studies reported that the cultivars NIBAM110 and NIBAM11 were highly susceptible, and the cultivars NIBAM109 and NIBAM108 moderately resistant to blast [7,44]. Because the growers who were aware of blast disease appeared to be less likely to have suffered significant damage to their crops from both blast and brown spot (possibly due to better control methods), we tested for an association between disease awareness and the cultivars grown by farmers. We found no correlation between blast awareness and cultivar selection, however, as all the NIBAM11 cv. farmers in Western Kenya were unaware of blast disease. Therefore, the low disease scores we observed for NIBAM11 cv. may be attributable to a lack of favorable conditions for the development of blast disease, because the infection rate of the cultivar was high. We could not obtain sufficient information to identify the factors that led to the lack of blast symptoms in NIBAM11 in our survey, and we therefore speculate that weather conditions might have led to poor disease development or that fungicides were used by some growers. 

The hybrid rice cultivar KEH10005 (Arize Tej Gold) showed a low incidence of brown spot and a low rate of infection by *C. miyabeanus*. To the best of our knowledge, this is the first report on brown spot resistance in KEH10005 cv., but the cultivar has been described as resistant to lodging and bacterial blight [45]. Given that we observed a marginal positive correlation between lodging and the occurrence of brown spot, it is likely that the low incidence and infection rate of this cultivar were due to its firm posture. Lodging can make crops vulnerable to fungal infection by bringing them closer to the ground, where humidity and higher inoculum concentrations can facilitate infection by fungal pathogens such as *C. miyabeanus*, which are known to survive in soil [46]. The observed low brown spot incidences and high infection rates of the cultivars NIBAM110 and NIBAM11 could be explained by previous fungicide sprays, unfavorable weather for disease development, or early sampling prior to disease development. The high incidences and infection rates of most rice cultivars with regard to brown spot disease indicate widespread susceptibility and an urgent need to breed for resistance in farmer-preferred rice cultivars, especially because some African cultivars that could serve as potential sources of brown spot resistance (e.g., NERICA 4 and NERICA 10) are available [47]. 

The visual disease scores in this study were not well correlated with actual fungal infection, as assessed by PCR-based diagnostics. However, this finding was expected because the accuracy of the identification of a disease based on symptoms requires expert prior knowledge, even though mixed symptoms can be also difficult to discern. To minimize the inaccuracy in symptom-based disease diagnosis, it has been proposed that assessments are always complemented by modern molecular biology techniques, such as DNA-based PCR, q-RT-PCR, or DNA barcoding, or the metagenomics of the infected plant tissue [48]. The use of molecular diagnostic tools can also overcome the complexity arising from multiple infections and, hence, mixed/overlapping disease symptoms. For foliar diseases, the multi-infection of rice leaf tissue can cause a mix of symptoms, such as blast, blight, mottling, scald, spots, or streaks [9,49]. To enhance rice disease surveillance, there is, therefore, a need to adopt accurate and inexpensive molecular diagnostic tools to complement the visual disease assessments in the field [1,50].

We endeavored to identify fungal pathogens within the leaf and neck tissues of rice through isolation in axenic culture and observed both pathogenic and endophytic species. It should be noted that we only scored phenotypically for blast and brown spot symptoms despite identifying several other putative pathogens. We speculate that some diseases not reported here are present on some farms because we isolated putative causal pathogens, such as *E. sorghinum*, which causes leaf spot, *M. albescens*, which causes leaf scald, and *Nigrospora oryzae*, a leaf spot and panicle branch rot pathogen [31,37,38,51]. If the widespread leaf spot symptoms observed in the fields in Kirinyaga were caused by *Nigrospora oryzae*, then this would be considered a new emerging threat to rice production in Kenya. Although we isolated and identified *N. oryzae* at a high frequency in rice leaves with widespread round spots, our study did not provide direct evidence for the fungus being the cause of the leaf spot by *N. oryzae*, because we were not able to fulfill Koch’s postulates [52]. Interestingly, some pathogens were isolated from parts of the plant where they are not known to cause symptoms, including seed rot fungi (*Alternaria padwickii* and *Aspergillus flavus*—a producer of carcinogenic aflatoxins) isolated from the neck tissue, sheath rot fungi (*Sarocladium oryzae*) from the leaves, glume blotch (*Pithomyces chartarum*) from the leaves and neck tissue, and head blight (*F. graminearum*) from the leaves. These cryptic infections at locations other than where symptoms normally occur have been reported previously [53]. We also observed the presence of non-pathogenic fungi, which could have influenced the disease symptoms [53,54]. Indeed, we isolated several non-pathogenic fungi in the current study that may play roles in maintaining plant health, protecting against abiotic and biotic stresses [54,55,56,57]. To enhance the surveillance of important diseases in rice, there is a need to develop a robust molecular diagnostic tool, such as an accurate multiplex PCR, to simultaneously detect multiple fungal species, because these methods are not affected by bottlenecks in microbial isolation [58]. Although this did not underestimate the essence of fungal species diversity in the rice tissues, we would like to acknowledge that the use of the ITS region in the identification of the fungal species may have reduced our accuracy in the identification of some species, e.g., *Aspergillus*, *Fusarium*, *Cladosporium*, and *Trichoderma*. A side-by-side comparison of the top five hits in NCBI and UNITE showed similar species, and we chose those that had been previously reported to attack rice or other related cereals as the putative species. Although it is generally recommended that the putative species that fall in this genus be re-tested using other, more effective, PCR primers, we believe the current results provide some insights into the species diversity in the tested rice tissues [59]. 

In summary, we obtained a snapshot of the resistance to brown spot and blast disease in cultivated rice in Kenya. This information is useful to breeders, because it may enhance cultivar deployment in farming communities, through the adoption of a specific breeding program based on the findings reported here, in the future. For example, we now consider NIBAM109 and NIBAM110 as potential resistant and susceptible checks, respectively, and used these cultivars in subsequent tests in the laboratory and at blast hotspot tests in Kenya. Information on pathogenic fungal species in rice across different agro-ecologies is also important because it can be used in the development of robust disease diagnosis tools to improve the quality of disease surveillance and management in sub-Saharan Africa. Finally, our study also highlights the enormous gap in the skills of agricultural extension officers in identifying rice diseases and recommends retraining and equipping agricultural extension agents to improve the quality of diagnostic services in the Kenyan rice value chain. Capacity building for agricultural extension agents can be achieved through the established PlantWise program, which trains plant doctors to operate at farm-level Plant Clinics, and which has been shown to be highly effective [1,60,61]. Our study provides further evidence in support of this conclusion, which is vital if smallholder farmers are to be provided with the necessary support to sustainably increase rice production in Kenya, which is urgently needed, but also very clearly possible with adequate systems in place.

## 4. Materials and Methods

### 4.1. Characteristics of the Study Sites

We conducted a survey during the rice-growing seasons of 2019 in Busia and Kisumu Counties (Western) and Kirinyaga County (Central), which are rice-growing regions in Kenya. Busia County is located at an elevation r1154–1239 m above sea level (asl), receives an annual rainfall of 760 mm to 2000 mm, and experiences temperatures ranging between 26 and 30 °C (maximum) and 14 and 22 °C (minimum) [62]. Within Busia County, the survey was conducted in Bunyala and Ng’elechom. Bunyala Irrigation Scheme lies 63 km to the southeast of Busia town at 0.1033° N, 34.0592° E and covers 1880 acres, which are cultivated by approximately 1500 farmers, who collectively produce 6524 tons of rice per year, according to recent figures. Ng’elechom lies about 16 km to the northeast of Busia town at 0.5644° N, 34.1588° E and has a combination of rain-fed and irrigated rice cultivation. Kisumu county lies at an elevation ranging 1134 m to 1525 m asl, with an annual rainfall ranging between 600 mm and 1630 mm and an annual temperature range of 25 °C to 35 °C (maximum) and 9 °C to 18 °C (minimum) [63]. Rice production in Kisumu County takes place in Ahero and West Kano Irrigation Schemes. Ahero is located about 23 km to the southeast of Kisumu city at 0.1744° S, 34.9203° E and occupies 4768 acres; it is farmed by about 2000 farmers, who recently produced 5896 tons of rice. West Kano is located 25 km to the south of Kisumu city at 0.1978° S, 34.8100° E and occupies 2830 acres under cultivation by 2200 farmers, who recently produced 5352 tons of rice per year. Kirinyaga county lies between 1158 and 5199 m above sea level, with rice production taking place in Mwea Irrigation Scheme, which is located within the low-altitude zones of the county. Mwea receives a mean annual rainfall ranging from 500 mm to 1250 mm, and its annual temperatures range from 16 to 18 °C (minimum) and 23 to 30 °C (maximum). The Mwea Irrigation Scheme is the largest and highest rice-producing region among the four schemes. It is located approximately 105 km north of Nairobi at 0.7329° S, 37.3478° E and covers 26,000 acres; it is farmed by 7022 farmers and produces 113,000 tons annually (County Government of Kirinyaga, 2020: https://kirinyaga.go.ke/kirinyaga-rice-farmers-receive-13-tons-of-seeds/ (accessed on 15 February 2022)). The scheme accounted for 78% of the irrigated area, 88% of production, and 98% of the gross value of rice production between 2005 and 2010 [64].

### 4.2. Study Design and Sampling

We interviewed, observed disease incidence/severity, evaluated key agronomic traits of the preferred cultivars, and collected rice samples from the three major rice-growing counties of Kenya. The study was conducted during the rice-grain ripening period (milk-mature stages) because these stages are critical in the observation of different blast symptoms [65,66]. Within the rice irrigation schemes, random farmers from different field blocks were recruited through the guidance of scheme leaders or agricultural extension workers. In Western Kenya, the survey was conducted on seventy-two farms, which were owned by fifty-five farmers. The samples collected depended on whether the symptoms of foliar or neck blast and/or brown spot were present (Table 1). Farms in Busia and Kisumu counties were surveyed between June and September, while the survey in Mwea, Kirinyaga County, was conducted in December 2019. In Mwea, Kirinyaga County, farmers provided information about preferred cultivars and reasons for their preference, and we collected rice leaves and neck tissues from twenty-six farmers who requested us to diagnose a blast-like disease in December 2019. We had planned to implement a similar study design in the three counties, but the implementation of the questionnaire in Kirinyaga was disrupted because it had been planned to take place in early 2020, when COVID-19 restrictions were implemented by the Kenyan government. During each field visit, farmers were provided with a description of the study and, thereafter, signed a letter of informed consent to participate. The final number of participant farmers was determined by willingness to participate, completion of the survey questionnaire and the availability of samples in the respective farms.

In each farm, trained research assistants and agricultural extension workers interviewed farmers regarding their preferred rice cultivars, their awareness of blast (whether they had heard about it and could identify it on their farms) and how they diagnosed, scored, and managed blast and brown spot disease. On farm visits, we assessed blast and brown spot incidences based on the percentage of rice plants exhibiting symptoms within three quadrats of 2 m^2^, which were assigned within a diagonal of a farmer’s field, up to a maximum of one hectare. The foliar blast severity was measured based on an estimate of leaf damage in the majority of the plants in a quadrat on a 0–9 scale, where 0 = no visible damage, 1 to 3 = varying degrees of hypersensitive reaction, and 4 to 9 = varying degrees of blast severity, as described previously [1,65]. Leaves and neck tissue were collected at random points within each farm. Asymptomatic and symptomatic tissue were collected and kept separately in individual sample bags. At least three sets of samples (symptomatic leaves and neck tissue, and asymptomatic leaves) were collected. Within each farm, samples were collected from plants representing each diagonal of the field. Agronomic traits assessed included the method of provision of water (rain-feeding or irrigation), plant height (cm), panicle color, crop vigor (increasing order of magnitude from 0 to 3), and lodging (0 to 3, with 0 being no lodging and 3 being where most plants were completely lodged). 

### 4.3. Sample Handling and Preparation

Samples were collected into brown paper bags, stapled, and transported in cooler boxes (Denver, Johannesburg, South Africa), and were stored at 4 °C prior to subsampling for plating, and at −20 °C for long-term storage or until they were subsampled for molecular analysis at the Plant Pathology Laboratory of the Biosciences eastern and central Africa—International Livestock Research Institute (BecA-ILRI) hub. Samples were split into two main subsets, as follows: those for use in the mycological characterization of fungal species were surface-sterilized and used to prepare plate cultures, while the remaining subset was freeze-dried and used for DNA extractions for molecular detection of two major fungi, *M. oryzae* and *C. miyabeanus*, using specific DNA primers. 

### 4.4. Molecular Detection of Magnaporthe oryzae and Cochliobolus miyabeanus in Rice

To complement visual assessment of blast and brown spot diseases, we used a PCR-gel electrophoresis method to test for the presence of causal pathogens in both asymptomatic and symptomatic leaves. Genomic DNA was extracted from the rice leaves using a CTAB method [67]. DNA quality and quantity were analyzed using 0.8% agarose gel electrophoresis and NanoDrop 2000C spectrophotometry (Thermo Fisher Scientific, Waltham, MA, USA). Genomic DNA samples were split into two subsets, for detection of each of the fungal species (*M. oryzae* and *C. miyabeanus*) in separate reactions. The blast pathogen was detected using pfh2a (5′-CGTCACACGTTCTTCAACC-3′) and pfh2b (5′-CGTTTCACGCTTCTCCG-3′) primer pair that amplified to a 687-base-pair region of the *Pot2* transposon [68]. The brown spot pathogen was detected using a cmSCD1-44F (5′-CATGTGTGCAGTAAAGTGACTC-3′) and cmSCD1-302R (5′-GTCTTGAGGAGGGGGTT-3′) primer pair that amplified to a 250-base-pair region of the unigene encoding *Scytalone dehydratase*, which is involved in fungal melanin biosynthesis [69]. The PCR was conducted in a reaction volume of 20 µL using a commercially available PCR PreMix (Bioneer corporation, Daejeon, South Korea), based on the manufacturer’s protocol. The reaction volume consisted of fungal DNA 3 µL (20 ng/µL), 0.4 µL of forward and reverse primers (10 pmol), and 16.2 µL of water topped up to into AccuPower Premix microwells. Thermocycler conditions were as follows: Initial denaturation for 3 min at 94 °C, followed by denaturation at 94 °C for 45 s, annealing at 55 °C (*M. oryzae*) or 60 °C (*C. miyabeanus*) for 45 s for a total of cycles, and a final extension at 72 °C for 45 s, for 10 min. Amplified products were fractionated by gel electrophoresis at 50 volts for 60 min in a 1.5% agarose gel, which was stained with GelRed (Biotium, Fremont, CA, USA). The gel was visualized under UV and samples with detectable bands were considered to have been infected by each of the specific pathogens. To confirm results, we applied Sanger DNA sequencing for a subset of samples with positive amplicons. For DNA sequencing, the amplicons were purified using a commercially available QIAquick PCR purification kit (Qiagen, Hilden, Germany). Purified samples were sequenced at the Segolip Unit of the ILRI Biosciences Genomic Platform using an ABI 3730 DNA Analyzer (Applied Biosystems, Waltham, MA, USA). To confirm the fungal species, sequences were compared with GenBank at the National Center for Biotechnology Information (NCBI).

### 4.5. Isolation and Identification of Diverse Fungi in Rice

We isolated diverse fungi from leaf and neck tissue of rice by direct plate culture. At least one asymptomatic and symptomatic leaf sample was used for all farms sampled in this study. In addition, we used representative samples of symptomatic neck rice tissue for isolations. Leaf and neck tissues were surface sterilized in 70% ethanol and rinsed twice in distilled water prior to plating in a moist 90-millimeter-diameter Petri dish. To maintain high relative humidity and moisture, two circles of 25-millimeter-diameter sterile Whatman filter paper (Whatman PLC, Maidstone, UK) were added to the bottoms of the Petri dishes and 5 mL of dH_2_O was added. Samples were propped with three sterile 1-milliter pipette tips above the filter paper within a capped Petri dish, sealed with Parafilm. Samples were incubated under white light conditions and a temperature of 28 °C. Morphological identification and isolation of distinct fungi commenced at 24 h after plating and continued for 96 h. Depending on morphological features of the fungal species, distinct morphotypes were obtained by single sporing re-isolation, using the isolation already established at the Mycology Platform of BecA-ILRI hub [70,71]. Pure cultures were multiplied on malt extract agar (MEA), amended with chloramphenicol, ampicillin and streptomycin to inhibit bacterial contamination, and incubated for 72 h in the dark at 28 °C prior to harvesting of mycelium for DNA extraction using an established protocol [72].

Fungal DNA was extracted from freeze-dried mycelium using a commercially available kit, following the manufacturer’s protocol (Zymo Research, Irvine, CA, USA). Quality and quantity of the DNA were assessed as described above. Fungal DNA was used in a PCR reaction with a universally conserved nuclear ribosomal internal transcribed spacer (ITS) region primer pairs ITS1-F (5′-CTTGGTCATTTAGAGGAAGTAA-3′) and ITS4 (5′-TCCTCCGCTTATTGATATGC-3′) for identification of each species [73]. PCR was carried in a 20-microliter reaction, with the following thermocycler conditions: an initial denaturation step at 94 °C for 4 min, followed by 35 cycles of denaturation at 94 °C for 45 s, annealing at 48 °C for 45 s and an extension at 72 °C for 45 s, and, subsequently, a final extension 72 °C for 10 min. The amplicons were analyzed using gel electrophoresis. Amplicons were purified using a QIAquick PCR purification kit and shipped for sequencing at Macrogen (Macrogen Europe B.V., Amsterdam, The Netherlands). To identify fungal species, a consensus of resultant sequences was generated in BioEdit ver. 7.2.5 [74] and compared with those in the NCBI database. Species names were assigned based on the highest percent homology, query size, and the most significant Expect value (E-value) in the Basic Local Alignment Search Tool (BLAST) output in NCBI.

### 4.6. Statistical Analysis

All statistical analyses were implemented in JMP Pro ver. 16 (SAS Institute Inc., Cary, NC, USA, 1989–2021). Descriptive statistics were conducted to show summaries/frequencies of responses for major predictor variables (county/sub-county, ecology, and cultivars) for the three rice-growing counties. Unlike Kirinyaga County, data from Busia and Kisumu survey utilized questionnaire fully completed following farmer interviews. We used the farm-level data from Busia and Kisumu to determine the incidence across counties and to statistically assess the factors that were associated with occurrence of blast and brown spot. A step-wise regression method was used to determine variables that best explained incidences of blast and brown spot, and severity of the former disease based on a model with a high coefficient of determination and low Akaike information criterion [75]. Analysis of variance was conducted for variables regarding occurrence/severity of the two diseases. Multiple means were compared using Tukey’s HSD at a significance level of 5%, while a comparison involving two means was based on Student’s *t*-test. A likelihood ratio test was implemented to compare proportions of samples with or without infection for either symptomatic or asymptomatic categories of rice tissue. Spearman’s non-parametric correlation tests were conducted to assess the associations between the three agronomic traits (i.e., the age of the crop, plant height, and lodging), and the incidence of the two diseases.

## Figures and Tables

**Figure 1 plants-11-01264-f001:**
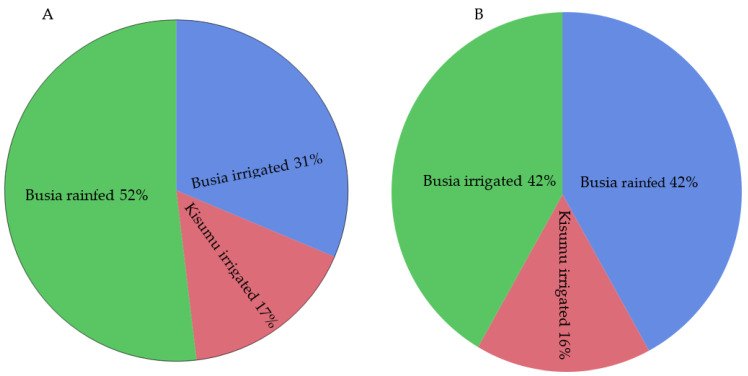
Incidence of blast (**A**) and brown spot (**B**) diseases in rice grown under different production systems in Busia and Kisumu counties of Kenya in 2019.

**Figure 2 plants-11-01264-f002:**
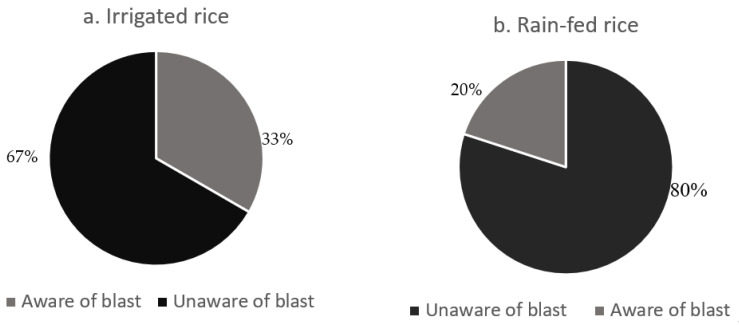
Blast awareness among the rice growers in irrigated and rainfed production systems in Busia and Kisumu counties of Kenya during 2019.

**Figure 3 plants-11-01264-f003:**
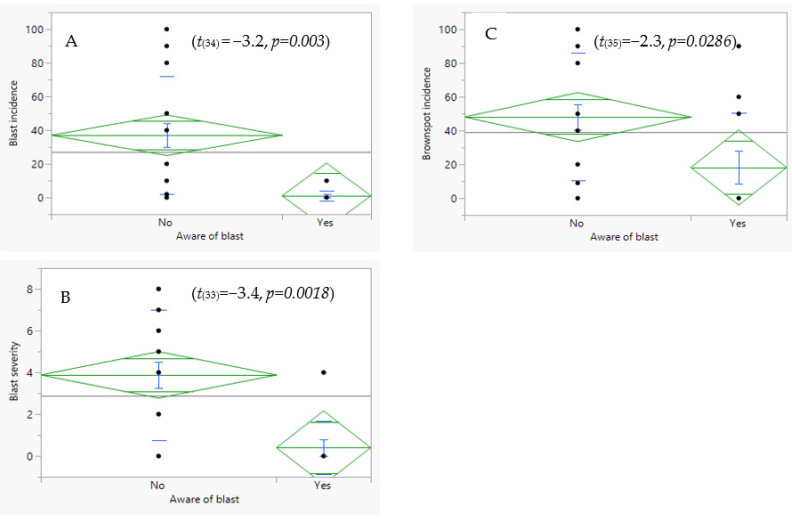
Association between growers’ awareness of rice blast disease symptoms and the incidence (**A**) and severity (**B**) of blast and incidence of brown spot (**C**) disease.

**Figure 4 plants-11-01264-f004:**
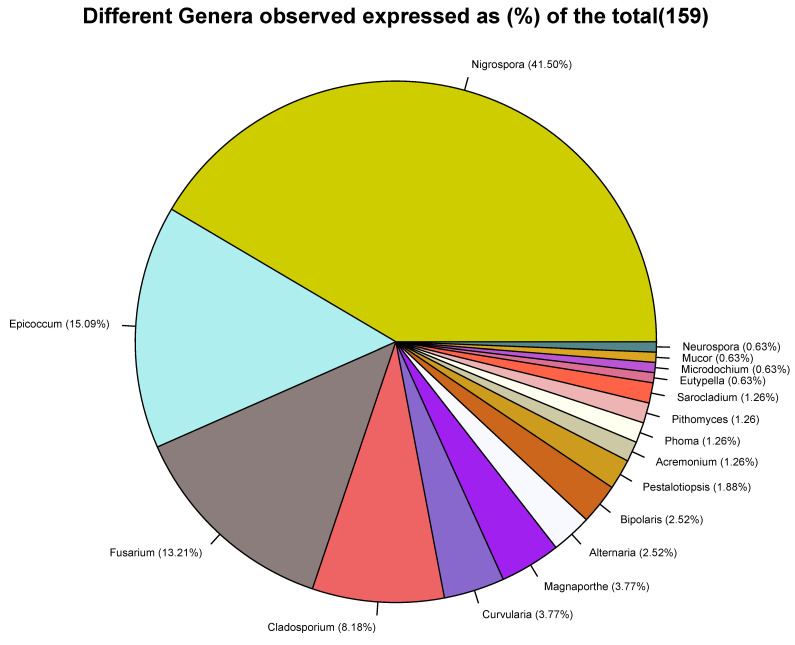
Occurrence of fungal genera in rice leaf and neck tissues collected from Busia, Kirinyaga, and Kisumu counties of Kenya surveyed in 2019.

**Figure 5 plants-11-01264-f005:**
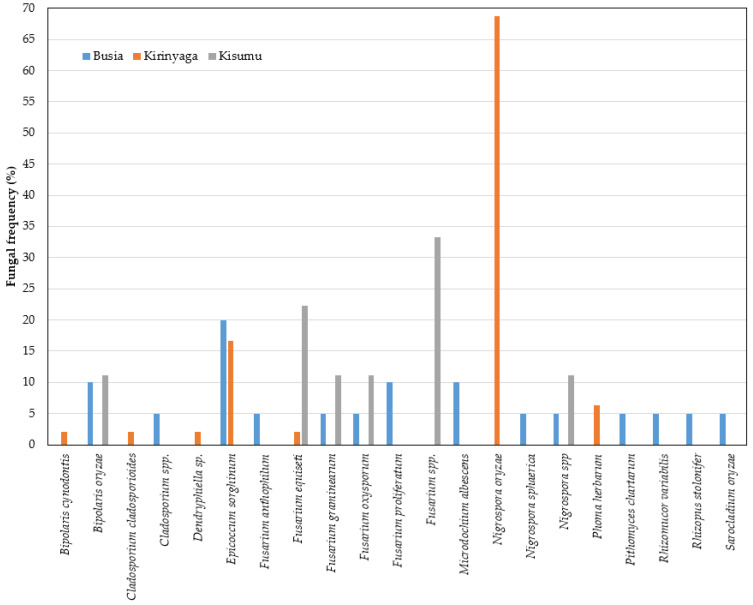
Occurrence of fungal species in rice leaf tissues from Busia, Kirinyaga, and Kisumu counties of Kenya surveyed in 2019.

**Figure 6 plants-11-01264-f006:**
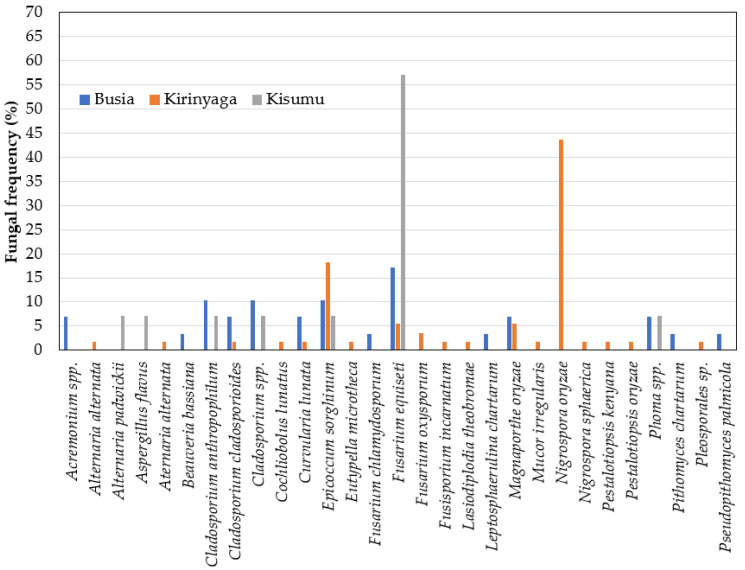
Occurrence of fungal species in neck tissues of rice from Busia, Kirinyaga, and Kisumu counties of Kenya surveyed in 2019.

**Table 1 plants-11-01264-t001:** Sites, irrigation types, farm sizes, number of farmers, and samples that were included in a survey conducted in the major rice areas of Kenya in 2019.

			Size of Farm (ha)			Farmers (n)	Rice Tissue Samples (*n*)		
							Leaves		Neck
County	Site	Irrigation Type	Minimum	Mean	Maximum		Asymptomatic	^¥^ Symptomatic	Symptomatic
Busia	Bunyala Central, Busia	Canal	0.04	0.34	0.65	20	20	32	7
	Ng’elechom, Busia	Canal	0.40	0.607	0.81	3	3	6	3
		Rain-fed	0.10	1.127	4.05	12	12	24	8
Kisumu	Ahero, Kisumu	Canal	0.40	0.903	3.24	15	11	23	6
	West Kano, Kisumu	Canal	0.40	0.549	0.81	13	10	18	2
Kirinyaga	Mwea East, Kirinyaga	Canal	0.10	0.751	4.05	19	8	19	19
	Mwea West, Kirinyaga	Canal	0.20	0.549	1.21	7	4	7	7

^¥^ Leaves were packaged and labelled separately, based on whether there were blast, brown spot, or mixed symptoms.

**Table 2 plants-11-01264-t002:** Percentages of farmers who preferred key traits of rice cultivars that were surveyed in Busia, Kirinyaga, and Kisumu Counties of Kenya in 2019.

County	Rice cultivar	Favorable Trait	Grower (n)	Growers (%)
Busia	Abednego from Uganda	Long grain and high yield	2	6
	KEH10004 (Arize 6444)	High tillering	1	3
	KEH10005 (Arize Tej Gold)	Less damage by pests and high grain yield	3	7
	NIBAM11 (Pishori)	Aroma and ready market	3	12
	NIBAM109 (BW196)	Tolerant to multiple stresses	3	9
	NIBAM108 (IR2793-80-1)	Resistant to multiple diseases	5	14
	NIBAM108 mixed with NIBAM109	These were the only cultivars whose seed was available for them.	1	3
	NIBAM110 (ITA310)	High grain yield	4	11
	IR05N221 (Komboka)	Aroma and high grain yield	3	9
	Pakistan K23 from Uganda	Drought tolerant and high grain yield	2	6
	Palata from Uganda	Aroma and high grain yield	1	3
	Vietnam from Uganda	High grain yield	6	17
subtotal			35	
Kisumu	KEH10005	-	2	7
	NIBAM11	-	2	7
	NIBAM108	-	9	32
	NIBAM110	-	7	25
	IR05N221	-	5	18
	Nyaboda from Uganda	High grain yield	2	7
	SUPA 1 from Uganda	High grain yield	1	4
subtotal			28	
Kirinyaga	NIBAM11	-	24	92
	IR05N221	-	1	4
	NIBAM109	-	1	4
subtotal			26	
Total			89	

**Table 3 plants-11-01264-t003:** Factors associated with the occurrence of rice blast and brown spot diseases in Busia and Kisumu Counties of Kenya during the crop seasons of 2019.

Source	DF	Sum of Squares	Mean Square	F Ratio	Prob > F
Blast Incidence					
Cultivar	9	7781.5505	864.617	2.2543	0.0525
Production system ^∞^	1	343.6142	343.614	0.8959	0.3529
Knowledge of blast	1	6553.9186	6553.919	17.0883	0.0004
County	1	431.4867	431.487	1.125	0.299
Model	12	20095.878	1674.66	4.3664	0.0009
Error	25	9588.333	383.53		
C. Total	37	29,684.211			
Blast Severity					
Cultivar	9	68.864343	7.65159	2.81	0.022
Production system	1	2.669446	2.66945	0.9803	0.3324
Knowledge of blast	1	42.862593	42.86259	15.7408	0.0006
County	1	14.643451	14.64345	5.3776	0.0296
Model	12	212.12039	17.6767	6.4916	<0.0001
Error	23	62.62961	2.723		
C. Total	35	274.75			
Brown Spot Incidence					
Cultivar	9	14,758.087	1639.787	3.1472	0.0106
Production system	1	1377.274	1377.274	2.6434	0.116
Knowledge of blast	1	5511.005	5511.005	10.5771	0.0032
County	1	6527.422	6527.422	12.5279	0.0015
Model	12	40,119.827	3343.32	6.4167	<0.0001
Error	26	13,546.839	521.03		
C. Total	38	53,666.667			

^∞^ Production system refers to whether the crop is rain-fed or irrigated.

**Table 4 plants-11-01264-t004:** Incidence and severity of blast and brown spot disease among the popular rice cultivars surveyed in Busia and Kisumu Counties of Kenya in 2019.

Cultivar	Growers (*n*)	Incidence (% ± SE)	Growers (*n*)	Severity (0–9) ± SE	^§^ Infection Rate (%)
Blast					
NIBAM11 (Pishori)	10	0 ± 9.4 C	10	0 B	50
Pakistan K23	3	0.4 ± 13.2 BC	3	1.5 ± 1.1 AB	33
Palata from Uganda	1	0.4 ± 13.2 ABC	1	1.5 ± 1.8 AB	0
Abednego from Uganda	2	5.8 ± 23.4 ABC	2	2.2 ± 2 AB	50
NIBAM109 (BW196)	5	8 ± 8 ABC	5	0.8 ± 0.8 B	80
IR05N221 (Komboka)	10	16.9 ± 7.7 ABC	10	2.8 ± 0.8 AB	40
Nyaboda from Uganda	2	19.6 ± 15.5 ABC	2	2.5 ± 1.3 AB	50
KEH10005 (Arize Tej Gold)	6	19.6 ± 15.5 ABC	6	2.5 ± 1.3 AB	67
NIBAM108 (IR2793-80-1)	14	21.9 ± 8.9 AB	14	3.2 ± 0.8 A	36
NIBAM110 (ITA310)	10	33.4 ± 10.1 AB	10	3.9 ± 0.9 A	80
Vietnam	7	35.4 ± 12 A	7	2.5 ± 1.0 AB	71
Brown Spot					
NIBAM11	10	16.7 ± 10.8 AB			70
Pakistan K23	3	21.4 ± 15.3 AB			67
Palata	1	61.4 ± 24.1 AB			100
Abednego	2	32.5 ± 27.3 AB			100
NIBAM109 (BW196)	5	34 ± 14 AB			60
IR05N221 (Komboka)	10	60.3 ± 9 A			70
Nyaboda	2	28.6 ± 17.9 AB			100
KEH10005 (Arize Tej Gold)	6	28.6 ± 17.9 AB			33
NIBAM108 (IR2793-80-1)	14	61.0 ± 10.3 A			64
NIBAM110 (ITA310)	10	7.8 ± 11.8 B			80
Vietnam	7	58.2 ± 12.1 AB			57

^§^ Infection rates are based on proportion of samples with detectable *C. miyabeanus* or *M. oryzae* using PCR. The cultivar KEH10004 (Arise 6444) was not included in the ANOVA because the research assistant omitted the disease score step by chance. Means followed by the same letter in a column do not differ significantly (Tukey’s HSD, alpha = 0.05). SE = standard error.

**Table 5 plants-11-01264-t005:** Fungal species identified in leaf and neck-tissue samples and their potential relationships with rice cultivation.

Fungal Species	Where Isolated in Rice	Effect on Rice	Reference
*Acremonium* spp.	neck	Potentially protective	[24]
*Alternaria padwickii*	neck	Seed rot	[25]
*Aspergillus flavus*	neck	Seed rot	[26]
*Beauveria bassiana*	neck	Potentially protective	[27]
*Cochliobolus miyabeanus*	leaf	Brown spot	[28]
*Cladosporium anthropophilum*	neck	No reported disease (NRD)	
*Cladosporium cladosporioides*	neck	NRD	
*Cladosporium* spp.	leaf and neck	NRD	
*Cochliobolus lunatus*	neck	Leaf spot, seedling blight and sheath rot	[29,30]
*Epicoccum sorghinum*	leaf and neck	Leaf spot	[31]
*Eutypella microtheca*	neck	NRD	
*Fusarium anthophilum*	leaf	NRD	
*Fusarium chlamydosporum*	neck	NRD	
*Fusarium equiseti*	leaf and neck	NRD	
*Fusarium graminearum*	leaf	Head blight	
*Fusarium oxysporum*	leaf	Seedling blight	[32]
*Fusarium proliferatum*	leaf	Bakanae disease	[33]
*Fusarium incarnatum*	neck	Spikelet rot	[34]
*Lasiodiplodia theobromae*	neck	Root rot complex disease	[35]
*Magnaporthe oryzae*	neck	Blast	[36]
*Microdochium albescens*	leaf	Leaf scald	[37]
*Mucor irregularis*	neck	NRD	
*Nigrospora sphaerica*	leaf	NRD	
*Nigrospora oryzae*	leaf	Leaf spot	[38]
*Pestalotiopsis kenyana*	neck	NRD	
*Pestalotiopsis oryzae*	neck	NRD	
*Phoma herbarum*	neck	Endophyte	
*Leptosphaerulina chartarum*	leaf and neck	Glume blotch	[39]
*Pseudopithomyces palmicola*	neck	NRD	
*Rhizopus stolonifer*	leaf	NRD	
*Sarocladium oryzae*	leaf	Sheath rot	[40]

NRD—No reported disease.

## Data Availability

The nucleotide sequences of the fungal isolates can be obtained at https://submit.ncbi.nlm.nih.gov/subs/?search=SUB10723570 (accessed on 15 February 2022).

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
