# Peer review of "Foliar Diseases and the Associated Fungi in Rice Cultivated in Kenya"

_plants, 2022, doi:10.3390/plants11091264_

Round 1

Reviewer 1 Report

Dear Authors,

Please find some questions/comments below. Additional suggestions are included in the Word (manuscript) file.

Abstract

  1. What is a cross-sectional survey?
  2. Probably it would be worth to mention that molecular identification of pathogens allows a better detection of pathogens (optional)

Keywords

  1. If endophytes are part of the keywords, the authors should also need to write a few words in the abstract.

Introduction:

  1. Please describe the symptomatology (how the lesions look like) of both blast and brown spot in rice
  2. Could you elaborate here what is known on disease management in the areas sampled?

M&M

  1. Ln 515-516. How many samples in total were collected? Please add that amount to the text.
  2. Please spell the gene name (SCD1) used for identification of C. miyabeanus
  3. Ln 604-607. Please clarify if you use stepwise regression or correlation or both (in that case add Spearman or the corresponding test that you used for assessing correlation among variables and also clarified when did the authors use the step-wise regression?)

Results

  1. Table 1 shows more farms that those indicated in text as sampled. Could you please clarify how those numbers were obtained? M&M: Ln. 494-496. If you are referring to another topic related to Table 2, please re-write clearly
  2. Table 1. Please fill names of “county” and “site/subcounty” to align with the rows (even when repeating them)
  3. What is the reason for which different numbers of farms were used in each county in the survey?
  4. Figure 1. The two circles would probably have to be inserted within a single figure (e.g., in TIFF) and added to the text.
  5. Could the authors please define what they mean by “ecologies” along the text?
  6. Please see the suggestion on Table 5 in the manuscript
  7. Were the samples frozen at -20C before pathogen isolation? I assume that they weren’t but please clarify because if they were, it could explain the low pathogen recovery. Otherwise please clarify that in M&M text.
  8. Co-occurrence of blast and brown spot: as written is confusing; are the authors indicating that even though the infection rates were higher in higher incidence or higher disease severity situations, the correlations did not indicate that. If so, please be specific.

Discussion

Overall, it is an interesting reading but it could be shortened.

Author Response

Thanks for these useful comments. We have gone through them and made major changes to reflect the issues raised. Please find he responses to each of the issues raised below:

Dear Authors,

Please find some questions/comments below. Additional suggestions are included in the Word (manuscript) file.

Abstract

  1. What is a cross-sectional survey? Because this term could not be understood by the reviewer, we thought it might hinder clarity to many readers. Therefore, we have deleted it and made some clerical changes in the sentence.  
  2. Probably it would be worth to mention that molecular identification of pathogens allows a better detection of pathogens (optional). We have emphasized on this in the discussion section.

Keywords

  1. If endophytes are part of the keywords, the authors should also need to write a few words in the abstract. We have deleted the term endophytes and rephrased the affected sentences in the new version of the manuscript.

Introduction:

  1. Please describe the symptomatology (how the lesions look like) of both blast and brown spot in rice: This has been effected. Thanks a lot for this comment.
  2. Could you elaborate here what is known on disease management in the areas sampled? We have added information on use of cultural methods such as clean beds and fungicide application. We have also acknowledged the prohibitory costs of fungicides and the lack of knowledge which may hinder appropriate application of the pesticides by small-scale farmers.

M&M

  1. Ln 515-516. How many samples in total were collected? Please add that amount to the text. This information has been provided in Table 1. We have also edited all affected sections in the manuscript, accordingly.
  2. Please spell the gene name (SCD1) used for identification of C. miyabeanus. We have provided this information.
  3. Ln 604-607. Please clarify if you use stepwise regression or correlation or both (in that case add Spearman or the corresponding test that you used for assessing correlation among variables and also clarified when did the authors use the step-wise regression?) We have clarified that the stepwise regression was used in identification of the factors that were statistically associated with the incidence and severity of the two diseases. We have also added Spearman correlation in the statistical analysis section of the materials and methods.

Results

  1. Table 1 shows more farms that those indicated in text as sampled. Could you please clarify how those numbers were obtained? M&M: Ln. 494-496. If you are referring to another topic related to Table 2, please re-write clearly. We have made changes in Table 1 and also reflected these changes in the text. 
  2. Table 1. Please fill names of “county” and “site/subcounty” to align with the rows (even when repeating them). The proposed changes were made. Thank you. 
  3. What is the reason for which different numbers of farms were used in each county in the survey? We have provided a clarifying statement in the materials and methods section that the final number of participant farmers was determined by willingness to participate, completeness of the survey questionnaire and the availability of samples in the respective farms.
  4. Figure 1. The two circles would probably have to be inserted within a single figure (e.g., in TIFF) and added to the text. We have grouped the figure. Should it be confusing, we can redraw it.
  5. Could the authors please define what they mean by “ecologies” along the text? We have replaced this with production systems and defined the latter.
  6. Please see the suggestion on Table 5 in the manuscript. We have accepted the proposed change.
  7. Were the samples frozen at -20C before pathogen isolation? I assume that they weren’t but please clarify because if they were, it could explain the low pathogen recovery. Otherwise please clarify that in M&M text. We have clarified in the materials and methods section that subsampling for plating was done prior to long-term storage of samples.
  8. Co-occurrence of blast and brown spot: as written is confusing; are the authors indicating that even though the infection rates were higher in higher incidence or higher disease severity situations, the correlations did not indicate that. If so, please be specific. This section has been rewritten for clarity. It should be noted that we did not collect data on brown spot severity, but incidence and infection rate. For blast we collected the severity, incidence and infection rate.

Discussion

Overall, it is an interesting reading but it could be shortened. We acknowledge your comments, as they have made us make key changes that have made the manuscript more clear. We edited and rephrased several sections based on these important contributions, particularly the comments within the pdf file.

Reviewer 2 Report

Text is too long, there are different problems discussed, but it was done superficially. I do not see necessity to discuss, for example, reasons for choosing on cultivars. 

I do not think, that data of one vegetation seasons’ survey during can give representative data about incidence and severity of diseases.

I am not sure, that sequencing of ITS region is sufficient to identify species, for example, species of Cladosporium, Microdochium. Normally, ITS region is suitable for identification of genus, and only for some species, but not for all mentioned species.

I do not agree with interpretation of data Lines 135 – 142. Development of diseases does not depend on awareness of farmers about diseases – it could be different technologies or different control measures, but not simply “awareness”.

Discussion is dedicated to situation in farms, it is not about obtained scientific data.

Author Response

Dear reviewer,

We sincerely appreciate your comments, which have enabled us to work towards improving the quality of this manuscript. We have done our best in responding to the concerns raised, and while some issues (e.g., multiple seasonal sampling) could not have been implemented, we have acknowledgment this as a weakness in the design while highlighting the important contribution of the current findings. We are thankful for your genuine criticism and we hope the current version of the manuscript will be acceptable for publication in Plants. Below find our specific responses to the issues raised. 

Issue 1: Text is too long, there are different problems discussed, but it was done superficially. I do not see necessity to discuss, for example, reasons for choosing on cultivars. We have trimmed some parts of the results. However, we could not remove the section on frequency of cultivars as this information is useful for our future efforts and for the regional breeders/policy.

I do not think, that data of one vegetation seasons’ survey during can give representative data about incidence and severity of diseases. We acknowledge that data from multiple crop seasons could add some value. However, we believe the differences between seasons could be less than the regional differences, particularly for the occurrence of fungal species. Given the constraints in the study design, we thought publishing this study would be valuable. Indeed, we could not implement the same design in one of the counties as it were in the other due to covid-19 restrictions by the government. It is for these reasons that we consider the value of the current findings to be more important than the weakness in design and request that the work be published in Plants. We are utilizing the findings in designing future experiments involving the amenable rice cultivars.

I am not sure, that sequencing of ITS region is sufficient to identify species, for example, species of CladosporiumMicrodochium. Normally, ITS region is suitable for identification of genus, and only for some species, but not for all mentioned species. Thanks for this comment. Indeed there are concerns about the accuracy of using ITS region for identification of some fungal species and also for assessing genetic diversity. However, ITS region has been generally accepted for basic fingerprinting of fungi (see Schoch et al 2012 and Badotti et al 2017).  Considering these counter arguments, we chose to screen the isolates using the primers for the region, as we also checked the quality of the sequences in comparison to the publicly available data in NCBI. We only gave specific names to sequences of fungi with high degree of homology and expected values within the database. We therefore expect a minimal error in reporting of the species.  Notwithstanding, we have made some significant changes in the manuscript in consideration of the concerns. We have provided an additional figure showing the percentage of observed fungal genera, and then considered the putative species in the rest of the figures and also revised to accommodate the concerns in the main text.

I do not agree with interpretation of data Lines 135 – 142. Development of diseases does not depend on awareness of farmers about diseases – it could be different technologies or different control measures, but not simply “awareness”. We acknowledge this concern and have rephrased to take care of the concerns, as the interpretation you raised is important. 

Discussion is dedicated to situation in farms, it is not about obtained scientific data. We attempted to identify key issues for discussion, while bringing together other interrelated aspects of the study. We believe this is what makes a story that is highlights the importance of the work to the stakeholders in the region and while highlighting key components of importance to the general scientific community. We have made changes in a few areas which we thought were not very clear to the readers.

Reviewer 3 Report

Paper reports an interesting research on the foliar diseases and the associated fungi in cultivated rice of Kenya. Results have a great utility value for the agricultural practice as well as breeders. It should be emphasized, that the paper is very well written, which shows very good organization, readability and grammar. In my opinion, the manuscript deserve to be published in Plants, an MDPI journal.

My general comments are as follows:

  1. Line 52 – reference [2] should be more recent if available.
  2. Line 53 – cited reference (IRRI 2018) should be moved to the chapter References and replaced by the following number [3].
  3. Line 112 – the information: (commonly known as Pishori) is repeated in line 115, it should be mentioned once.
  4. Lines 269 - 273 – sentence should be corrected
  5. Lines 310 - 315 – Latin names of fungi should be written in italics

Author Response

Paper reports an interesting research on the foliar diseases and the associated fungi in cultivated rice of Kenya. Results have a great utility value for the agricultural practice as well as breeders. It should be emphasized, that the paper is very well written, which shows very good organization, readability and grammar. In my opinion, the manuscript deserve to be published in Plants, an MDPI journal.

Dear Reviewer,

We sincerely appreciate your assessment and comments. Below find our responses to the issues raised. In addition, we made changes which we believe have improved the quality of the manuscript.

My general comments are as follows:

  1. Line 52 – reference [2] should be more recent if available. We have replaced this reference.
  2. Line 53 – cited reference (IRRI 2018) should be moved to the chapter References and replaced by the following number [3]. This reference has also been revised and the text has been rephrased.
  3. Line 112 – the information: (commonly known as Pishori) is repeated in line 115, it should be mentioned once. We have addressed this concern.
  4. Lines 269 - 273 – sentence should be corrected. We corrected this sentence by deleting the misplaced text.
  5. Lines 310 - 315 – Latin names of fungi should be written in italics. This has been implemented throughout the manuscript.

Round 2

Reviewer 2 Report

Authors have explained their consideration, but have not change significantly text. There are many different information, part of this information has practical, but no significant importance - for example Table 2. I do not agree with interpretation of results related to "awareness of diseases". Awareness of disease does not change anything, actions are necessary. ITS region is not suitable for identification of all species, for example Alternaria can be identified only to genera level, not species.

Author Response

Dear Reviewer:

We sincerely acknowledge your positive criticism. 

In response to the issues raised, we have added some sentences in the discussion section. Please following sections: 

Page 14: Lines 356-362

Page 16: Lines 454-459

We hope these changes will enhance clarity of the manuscript.
